# Identification of natural cytochalasins as leads for neglected tropical diseases drug discovery

**Marilia Valli** [1]*, **Julia Medeiros Souza**[1], **Rafael Consolin Chelucci**[1], **Carolina Rabal Biasetto**[2], **Angela Regina Araujo**[2], **Vanderlan da Silva Bolzani**[2], **Adriano Defini Andricopulo**[1]*

**1** Laboratory of Medicinal and Computational Chemistry (LQMC), Centre for Research and Innovation in Biodiversity and Drug Discovery (CIBFar), São Carlos Institute of Physics (IFSC), University of São Paulo (USP), São Carlos, SP, Brazil, **2** Nuclei of Bioassays, Biosynthesis and Ecophysiology of Natural Products (NuBBE), Department of Organic Chemistry, Institute of Chemistry, São Paulo State University (UNESP), Araraquara, SP, Brazil

* aandrico@ifsc.usp.br (ADA); marilia.valli@ifsc.usp.br (MV)

## Abstract

Investigating the chemical diversity of natural products from tropical environments is an inspiring approach to developing new drug candidates for neglected tropical diseases (NTDs). In the present study, phenotypic screenings for antiprotozoal activity and a combination of computational and biological approaches enabled the identification and characterization of four cytochalasins, which are fungal metabolites from Brazilian biodiversity sources. Cytochalasins A-D exhibited $IC_{50}$ values ranging from 2 to 20 μM against intracellular *Trypanosoma cruzi* and *Leishmania infantum* amastigotes, values comparable to those of the standard drugs benznidazole and miltefosine for Chagas disease and leishmaniasis, respectively. Furthermore, cytochalasins A-D reduced *L. infantum* infections by more than 80% in THP-1 cells, most likely due to the inhibition of phagocytosis by interactions with actin. Molecular modelling studies have provided useful insights into the mechanism of action of this class of compounds. Furthermore, cytochalasins A-D showed moderate cytotoxicity against normal cell lines (HFF-1, THP-1, and HepG2) and a good overall profile for oral bioavailability assessed in vitro. The results of this study support the use of natural products from Brazilian biodiversity sources to find potential drug candidates for two of the most important NTDs.

## Introduction

Neglected tropical diseases (NTDs) affect 1.7 billion people worldwide, mostly in tropical and subtropical areas, and they have high morbidity and mortality rates [1, 2]. Major global initiatives, such as the road map for NTDs 2021–2030, have been created to set global targets and milestones to prevent, control, eliminate or eradicate the set of 20 diseases and attain the United Nations Sustainable Development Goals [1]. Despite worldwide efforts to develop new therapies for these diseases, no new chemical entities (NCEs) have been developed for NTDs in the 21st century [3, 4]. During this period, the few approved products were repurposed

**Data Availability Statement:** All relevant data are within the article.

**Funding:** This work was supported by the Fundação de Amparo à Pesquisa do Estado de São

Paulo (FAPESP, https://fapesp.br/) grants #2013/07600-3 (CIBFar-CEPID), #2014/50926-0 (INCT BioNatCNPq/FAPESP), Conselho Nacional de Desenvolvimento Científico e Tecnológico (CNPq, https://www.gov.br/cnpq/pt-br) and Coordenação de Aperfeiçoamento de Pessoal de Nível Superior (CAPES, https://www.gov.br/capes/pt-br) for grant support and research fellowships. The authors acknowledge the scholarships conferred to MV (Fapesp #2019/05967-3) and JM (Fapesp #2019/06034-0). The funders had no role in study design, data collection and analysis, decision to publish, or preparation of the manuscript.

**Competing interests:** The authors have declared that no competing interests exist.

compounds, drug combinations, biologicals or new formulations. Remarkably, the vast majority of available drugs present one or more limitations, including low efficacy, toxicity problems, drug resistance, duration of treatment and costs [2]. Therefore, new treatments are urgently needed.

Natural products from tropical environments and their structural analogues represent promising sources for the expansion of chemical space for NTD drug discovery [5]. The potential use of unexplored compounds from nature as innovative drugs remains very attractive, especially in a country such as Brazil, which is particularly rich in biodiversity and therefore a source of chemical diversity [6–8].

In this study, we employed a combination of computational and biological approaches to identify new scaffolds, starting with the screening of natural product libraries from Brazilian organisms [9, 10]. In vitro phenotypic assays and pharmacokinetics evaluation as well as cheminformatics and molecular modelling tools were used to investigate a series of bioactive compounds with antiparasitic properties.

## Materials and methods

### Screening a collection of natural products

A screening of a natural product collection was performed by using cell assays for infection with *Trypanosoma cruzi* and *Leishmania infantum*. The collection includes 320 compounds from the metabolic classes terpenes, alkaloids, peptides and amides that were isolated by plant species or endophytic fungi. For further evaluation in this study, cytochalasins A-D (CAS numbers: CytA: 14110-64-6, CytB: 14930-96-2, CytC: 22144-76-9, CytD: 22144-77-0) were obtained from Sigma-Aldrich (St. Louis, MO, USA). The purity of these compounds was ≥97%. The stock solution of the compounds was prepared in 100% DMSO (dimethyl sulfoxide—Sigma-Aldrich).

### Cell culture

THP-1 (human monocyte cell line) cells were cultivated in RPMI medium supplemented with 10% foetal calf serum (FCS) (Cultilab, Campinas, SP, BR) and 1% penicillin and streptomycin (pen/strep—Sigma-Aldrich) at 37˚C under 5% $CO_2$, and HFF-1 (human fibroblast) and HepG2 (hepatocytes) cells were cultivated in DMEM without phenol red (Cultilab) supplemented with 10% FBS (Sigma-Aldrich) and 1% pen/strep at 37˚C under 5% $CO_2$.

### Parasite cultures

The epimastigote forms of the *T. cruzi* Tulahuen strain, which expresses the *E. coli* β-galactosidase gene lacZ [11], were grown in liver infusion tryptone (LIT) supplemented with 10% FCS (Cultilab) and 1% pen/strep (Sigma–Aldrich) at 28˚C. Metacyclogenesis from epimastigotes to trypomastigotes was induced by the incubation of the epimastigotes in Grace's insect medium (Sigma-Aldrich, 28˚C) every 5 days. After 5 days, the trypomastigote forms were used to infect the HFF-1 cells. The promastigote forms of *L. infantum* were maintained in Schneider's insect medium (Sigma-Aldrich) supplemented with 10% FCS (Cultilab) and 1% pen/strep (Sigma-Aldrich) at 28˚C and subcultured every 6 or 7 days.

### *T. cruzi* intracellular amastigote assays

In vitro assays against *T. cruzi* were performed as described previously [12]. HFF-1 human fibroblasts were seeded at $4x10^3$/well in 96-well tissue culture plates in DMEM (Cultilab) without phenol red and then incubated overnight (37˚C, 5% $CO_2$). Next, trypomastigotes were

added at $4x10^4$/well, and the plates were incubated (37˚C, 5% $CO_2$). After 24 h, 2-fold serial dilutions of the test compounds were added at concentrations ranging from 64 to 0.12 µM, and the plates were then incubated (37˚C, 5% $CO_2$). Each compound concentration was evaluated in duplicate. All the plates included BZ (64–0.12 µM) (Sigma-Aldrich) as a positive control and untreated wells (100% growth) as a negative control. After 120 h, chlorophenol red β-D-galactopyranoside (CPRG, Sigma-Aldrich) was added to each well. The absorbance was measured at 570 nm in an automated SpectraMax 384 microplate reader (Sunnyvale, CA, USA), and the data were analysed using GraphPad Prism version 8.0 (San Diego, CA, USA) for $IC_{50}$ value calculation.

## *L. infantum* intracellular amastigote assays

In the intracellular amastigote assay, THP-1 cells were seeded at $2x10^4$/well (RPMI-1640, 100 µL) with phorbol 12-myristate 13-acetate (PMA) at 20 ng/mL for differentiation into macrophages. After incubation for 72 h (5% CO2, 37˚C), the medium was aspirated, and late-stage *L. infantum* promastigotes were added ($2x10^5$/well, 100 µL). Following 24 h of incubation, the medium was aspirated to clear extracellular parasites, and the compounds were added in serial dilutions (2-fold) to final concentrations of 64–0.12 µM. The plates included negative controls (100% growth) and MIL (64–0.12 µM) as a positive control. After 120 h of incubation, the medium was removed, and the cells were fixed in methanol and stained with Giemsa. The average number of intracellular amastigotes per THP-1-cell was determined using an inverted microscope and a cell counter. Growth inhibition is expressed as a percentage of the average number of amastigotes per macrophage in the negative control wells. The $IC_{50}$ values were obtained using GraphPad Prism version 8.0.

## Cytotoxicity in HFF-1 and HepG2 cell lines

The cytotoxicity of the compounds against HFF-1 fibroblasts and HepG2 hepatocytes was evaluated using the MTS tetrazolium assay (Promega, Madison, WI, USA) [13]. HFF-1 fibroblasts were seeded at $4x10^3$/well in 96-well culture plates in DMEM without phenol red and incubated overnight (37˚C, 5% $CO_2$). HepG2 hepatocytes were seeded at $7x10^3$/well in 96-well culture plates in DMEM (Cultilab) with supplements (10% FCS, 1% pen/strep) and incubated overnight (37˚C, 5% $CO_2$). The compounds were added in 2-fold serial dilutions at the same concentrations described before, and the plates were incubated as described before. Each compound concentration was evaluated in duplicate. All the plates included doxorubicin (DOX) (Sigma-Aldrich) as a positive control (10–0.01 µM) and untreated wells as a negative control (100% growth). After 72 h, 15 µL of MTS (Promega) was added, and the plates were incubated for 4 h. The absorbance was measured at 490 nm in an automated SpectraMax 384 microplate reader (Sunnyvale, CA, USA), and the data were analysed using GraphPad Prism version 8.0 for $CC_{50}$ value calculation. The percentage of nonviable cells was determined and compared to the negative control wells (100% growth).

## Cytotoxicity in the THP-1-cell line

THP-1 cells were seeded at $2x10^4$/well in 96-well culture plates in RPMI 1640 supplemented (10% FCS, 1% pen/strep) and incubated overnight (37˚C, 5% $CO_2$). The compounds were added in 2-fold serial dilutions (64–0.12 µM), and the plates were incubated as described before. Each compound concentration was evaluated in duplicate. All the plates included DOX (Sigma-Aldrich) as a positive control (10–0.01 µM) and untreated wells as a negative control (100% growth). After 72 h, 20 µL of resazurin (Sigma-Aldrich) was added, and the plates were incubated for 4 h. The absorbance was measured at 536 and 588 nm in an

automated microplate reader SpectraMax Gemini, and the data were analysed using GraphPad Prism version 8.0 for $CC_{50}$ value calculation. The percentage of nonviable cells was determined and compared to the negative control wells (100% growth).

## Antiphagocytic activity

The antiphagocytic assay was performed with THP-1 cells seeded at $2x10^4$/well (RPMI-1640, 100 μL) with phorbol 12-myristate 13-acetate (PMA) at 20 ng/mL for differentiation into macrophages. Following 72 h of incubation (5% CO2, 37˚C), the medium was aspirated, and compounds were added in serial dilutions (2-fold) to reach final concentrations of 20–2.5 μM and incubated for 2 h. After incubation, two experiments were performed. 1) The compounds were removed, and late-stage *L. infantum* promastigotes were added ($4x10^5$/well, 100 μL). 2) The compounds were not removed, and late-stage *L. infantum* promastigotes were added ($4x10^5$/well, 100 μL). These plates included negative controls (maximum infection without inhibitors). After 24 h of incubation, the medium was removed, and the cells were fixed in methanol and stained with Giemsa. The methodology was adapted from the literature [14, 15]. The average number of infected THP-1 cells was determined using an inverted microscope and a cell counter. Phagocytosis inhibition is expressed as a percentage of infected cells determined by counting 200 random cells.

## Leishmanicidal assay using *L. infantum* promastigotes

Promastigotes of *L. infantum* ($1x10^6$ parasites/well) were distributed in 96-well plates, and the test compounds (cytochalasins A-D) were added at concentrations of 20–2.5 μM. The plates were incubated under the same conditions as previously described for 24 h. The leishmanicidal activity was determined by adding 20 μL of resazurin (Sigma-Aldrich), and the plates were incubated for 4 h. The absorbance was measured at 536 and 588 nm in an automated SpectraMax Gemini microplate reader, and the data were analysed using GraphPad Prism version 8.0 for the $CC_{50}$ value calculation. The percentage of nonviable parasites was determined and compared to the negative control wells.

## Statistical analysis

The analyses were performed using GraphPad Prism version 8.0 (GraphPad Software, San Diego, CA, USA). Both the $IC_{50}$ and $CC_{50}$ values were calculated by adjusting the sigmoid dose-response curves, and the selectivity index (SI) values were determined by $CC_{50}/IC_{50}$ [16]. Statistical analyses were performed using one-way ANOVA followed by Dunnett's multiple comparison test.

## Molecular docking of cytochalasins

Mol2 files of the docked compounds were generated using the molecular modelling software package Sybyl-X 2.1.1 (Certara, St. Louis, MO, USA), which was run on CentOS Linux workstations. The 3D conformational energy was minimized using the Tripos force field and Powell's method [17]. Partial atomic charges were calculated using the Gasteiger-Hückel method, and the molecules were considered to be in an implicit aqueous environment (dielectric constant of 80.0). The molecules were docked with the program GOLD 2020 (Cambridge Crystallographic Data Centre, Cambridge, UK) with the actin structure (PDB ID 3EKS, 1.80 Å) [18]. The target preparation consisted of removing all crystallographic waters, ligand (CY9), and ATP molecules and adding hydrogens. The binding site was defined as a sphere with a 10 Å radius centred on the crystallographic ligand (CY9). The compounds were docked

by applying the default GOLD parameters and using the ChemScore scoring function. An examination of the docking results was performed with PyMOL 1.7.x (Schrödinger, New York, NY).

### In silico ADME property determination

The in silico pharmacokinetic properties were calculated with the Swiss ADME Web Tool by considering the following parameters: the clog$\underline{P}$, size, polarity, unsaturation (fraction of Csp$^3$), flexibility (nRotb), solubility, BBB permeation, CYP enzymes and metabolism, Lipinski's Rule of 5 and drugability, and PAINS (pan-assay interference compounds) alerts [19, 20].

### Experimental determination of the distribution coefficient (Log$D$7.4)

To analyse cytochalasin lipophilicity, a methodology based on the retention time of molecules in stationary phase was used in LC-MS/MS (liquid chromatography-tandem mass spectrometry) Prominence UFLC (Shimadzu Corporation, Kyoto, Japan) and an interface with an LCMS-8045 triple quadrupole mass spectrometer (Shimadzu Corporation, Kyoto, Japan) with an electrospray ionization source (ESI). The chromatogram was obtained using a Supelco Ascentis express RP amide HPLC column (5 cm x 2.1 mm, 2.7 μM). The mobile phases consisted of 5% methanol in 10 mM ammonium acetate pH 7.4 (A) and 100% methanol (B). The mobile phase was eluted in binary gradient mode, and the gradient was as follows: 0 min, 95% A; 0.3 mins. 100% A; 5.2 mins. 0% A; 5.6 mins. 0% A; 5.8 mins. 100% A; 7.0 mins. And 100% A. The run time was 7 minutes, and the sample injection volume was 5 μL. Test compounds were prepared at 1.0 mg/mL by adding the stock solution at (1:1) mobile phases A:B (internal standard at 200 nM), and the DMSO concentration was lower than 2%. The lipophilicity of the compounds was assessed by individually injecting the test compounds and a series of eight commercial drugs, covering a Log$D$ range of -1.86 to 6.1 (acyclovir -1.86, atenolol 0.16, antipyrine 0.38, fluconazole 0.50, metoprolol 1.88, ketoconazole 3.83, tolnaftate 5.40, and amiodarone 6.10) [21–23]. The retention time (in minutes) of each of the eight standards was plotted against their Log$D$ values. The resulting equation for the calibration curve (y = mx + b) was used to calculate the Log$D$ values for the cytochalasins.

### Human and mouse liver microsomal stability assay

The metabolic stability of the cytochalasins was evaluated in human pooled liver human microsomes (20 mg/ml, GIBCO) and pooled liver CD1 mouse microsomes (20 mg/ml, GIBCO). Test cytochalasins were prepared at a concentration of 0.5 μM and incubated with 0.25 mg/mL liver microsomes at pH 7.4), and the DMSO concentration was lower than 1%. The reaction was started by adding the cofactor NADPH at 0.5 μM. Samples were taken at time point 0 (immediately following the addition of the co-factor), 5, 10, 20, 30 and 60 minutes. The reaction was stopped by adding quench solution containing acetonitrile:methanol (1:1) (internal standard tolbutamide at 50 nM). The samples were centrifuged (3500 rpm/30 min.) to obtain the pellet of precipitated microsomal protein. The supernatant fractions were quantified by LC-MS/MS (liquid chromatography-tandem mass spectrometry) Prominence UFLC (Shimadzu Corporation, Kyoto, Japan) and interfaced with an LCMS-8045 triple quadrupole mass spectrometer (Shimadzu Corporation, Kyoto, Japan) with an electrospray ionization source (ESI). The peak area ratios (analyte/internal standard) were converted to the % remaining using the area ratio at time 0 as 100%. The half-life ($t1/2 = \ln(2)/k$) in minutes and intrinsic clearance ($Clint = k \times 1000/(0.25)$) in μL/min/mg were calculated using a nonlinear regression from the % remaining versus the incubation time. From this plot, the slope (k) was determined. The chromatogram for analysis was obtained on a Supelco Ascentis express C18

column (3 cm x 2.1 mm, 5 μM). The mobile phases consisted of water + 0.1% formic acid (A) and acetonitrile + 0.1% formic acid (B). The mobile phase was eluted in binary gradient mode, and the gradient was as follows: 0 min, 95% A; 0.05 mins. 95% A; 0.3 mins: 2% A; 0.7 mins: 2% A; 0.8 mins: 95% A; 1.15 mins 95% A; and 2.0 mins 95% A. The run time was 2 minutes, the sample injection volume was 10 μL, and the flow rate was 0.7 mL/min.

## Parallel artificial membrane permeability assays (PAMPA)

A parallel artificial membrane permeability assay was performed in a 96-well precoated PAMPA plate system (Corning Gentest # 353015). The compound solutions were prepared by diluting the stock solutions (10 mM) in phosphate-buffered saline (PBS) pH 6.5 at a final concentration of 10 μM and then added to the donor portion of the plate (300 μl/well), while PBS pH 7.4 (200 μl/well) was added to the acceptor portion, and the DMSO concentration was lower than 1%. The two portions of the plate were then coupled, and the system was incubated for 5 h at 37°C. Samples of the initial donor solution (T0) were collected and stored at -20°C. At the end of the incubation, samples were collected from the donor and acceptor plates and then added to plates containing quench solution (10% water and 90% methanol:acetonitrile (50:50) + 50 nM tolbutamide). The T0 samples were treated similarly. The final concentrations of compounds in the donor, acceptor and T0 wells were quantified by LC-MS/MS (liquid chromatography-tandem mass spectrometry) Prominence UFLC (Shimadzu Corporation, Kyoto, Japan) and interfaced with an LCMS-8045 triple quadrupole mass spectrometer (Shimadzu Corporation, Kyoto, Japan) with an electrospray ionization source (ESI). The chromatogram for analysis was obtained on a Supelco Ascentis express C18 column (3 cm x 2.1 mm, 5 μM). The mobile phases consisted of water + 0.1% formic acid (A) and acetonitrile + 0.1% formic acid (B). The mobile phase was eluted in binary gradient mode, and the gradient was as follows: 0 min, 95% A; 0.05 mins. 95% A; 0.3 mins: 2% A; 0.7 mins: 2% A; 0.8 mins: 95% A; 1.15 mins 95% A; and 2.0 mins 95% A. The run time was 2 minutes, the sample injection volume was 10 μL, and the flow rate was 0.7 mL/min. The results were used to calculate an effective permeability (Pe) value.

## Kinetic solubility

To determine the kinetic solubility, 10 mM samples of each cytochalasin were transferred to a 96-well plate (incubation plate) in duplicate; for each sample on the plate, 195 μL of PBS buffer pH 7.4 and 2.0 (final concentration of 250 μM) was added, and the DMSO concentration was 2.5%. The plate was sealed and shaken for 24 hours (200 rpm, r.t.). The precipitates on the incubation plate were removed by centrifugation (15 min., 3000 rpm, r.t.). The supernatant fractions were quantified by LC-MS/MS (liquid chromatography-tandem mass spectrometry) Prominence UFLC (Shimadzu Corporation, Kyoto, Japan) and interface with an LCMS-8045 triple quadrupole mass spectrometer (Shimadzu Corporation, Kyoto, Japan) with an electrospray ionization source (ESI). From a 10 mM standard solution, an intermediate standard dilution of 0.5 mM in a 1:1 acetonitrile:water solution was prepared. Calibration curves were prepared for each cytochalasin by diluting the intermediate standard several times to reach the desired concentrations of 50, 40, 20, 2, and 1 μM. The resulting equation for the calibration curve (y = mx + b) was used to calculate the experimental concentration values for the cytochalasins. The chromatogram for analysis was obtained on a Supelco Ascentis express C18 column (3 cm x 2.1 mm, 5 μM). The mobile phases consisted of water + 0.05% formic acid (A) and acetonitrile + 0.05% formic acid (B). The mobile phase was eluted in binary gradient mode, and the gradient was as follows: 0 min, 98% A; 1.2 mins. 2% A; 2.0 mins: 2% A; and re-equilibration

time: 0.6 min., 98% A. The run time was 2 minutes, the sample injection volume was 5 μL, and the flow rate was 0.6 mL/min.

## Results

The biological screening of a collection of natural products against *T. cruzi* and *L. infantum*, consisting of piperamide and piperidine derivatives, cyclopeptides and terpenes, among others, revealed a small series of bioactive cytochalasins, as shown in Fig 1. Cytochalasins A-D are natural products in the terpene metabolic class produced by fungi in the *genera Penicillium*, *Aspergillus*, *Xylaria* and *Phomopsis* and possess a macrocyclic ring of 14 (cytochalasins A and B) - 11 (cytochalasins C and D) members, a carbonyl (cytochalasin A) or hydroxy (cytochalasin B) substituent at position C20, and a double bond position at C5-C6 (cytochalasin C) or C6-C12 (cytochalasin A, B and D).

Based on the preliminary encouraging results from the biological evaluation, these cytochalasins were selected with the goal of investigating and characterizing their potency, pharmacokinetic properties, selectivity index and mechanism of action. The results of the evaluation on the trypanocidal and leishmanicidal properties of cytochalasins A-D are depicted in Table 1. Cytochalasin A showed significant potency against the intracellular amastigote forms of *T. cruzi* and *L. infantum*, with $IC_{50s}$ of 3.02 μM and 2.76 μM, respectively. Cytochalasins B, C and D exhibited $IC_{50}$ values of 16.87, 10.24 and 20.15 μM for *T. cruzi* and approximately 22–23 μM

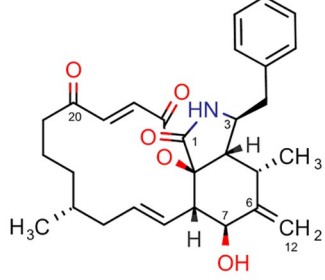

**Cytochalasin A**
*T. cruzi* inhibition = 99.7%
*L. Infantum* inhibition = 100%

**Cytochalasin B**
*T. Cruzi* inhibition = 79.4%
*L. Infantum* inhibition = 57.7%

**Cytochalasin C**
*T. Cruzi* inhibition = 72.8%
*L. Infantum* inhibition = 58.5%

**Cytochalasin D**
*T. Cruzi* inhibition = 69.5%
*L. Infantum* inhibition = 64.9%

**Fig 1. Naturally occurring cytochalasins A-D and their percentage inhibition in the screening against *T. cruzi* and *L. infantum* corresponding to a compound concentration of 50 μM.**

**Table 1. Trypanocidal and leishmanicidal activities of cytochalasins A–D against intracellular *T. cruzi Tulahuen LacZ* and *L. infantum* amastigotes and cytotoxicity for fibroblasts (HFF-1), human macrophages (THP-1) and hepatocytes (HepG2).**

| | *T. cruzi* IC$_{50}$ (μM ± SD) | HFF-1 CC$_{50}$ (μM ± SD) | SI$^1$ | *L. infantum* IC$_{50}$ (μM ± SD) | THP-1 CC$_{50}$ (μM ± SD) | SI$^2$ | HepG2 CC$_{50}$ (μM ± SD) | SI$^3$ | SI$^4$ |
|---|---|---|---|---|---|---|---|---|---|
| Cytochalasin A | 3.02 ± 2.16 | 27.07 ± 0.17 | 8.96 | 2.76 ± 1.17 | 17.03 ± 1.34 | 6.17 | 21.12 ± 4.51 | 6.99 | 7.65 |
| Cytochalasin B | 16.87 ± 1.47 | 21.81 ± 3.57 | 1.29 | 23.87 ± 3.02 | 109.20 ± 2.4 | 4.57 | 46.27 ± 19.33 | 2.74 | 1.93 |
| Cytochalasin C | 10.24 ± 1.13 | 76.3 ± 3.12 | 7.45 | 23.22 ± 4.90 | 84.84 ± 6.84 | 3.65 | 22.46 ± 6.47 | 2.19 | 0.96 |
| Cytochalasin D | 20.15 ± 3.27 | 96.47 ± 10 | 4.78 | 22.54 ± 0.97 | 122.20 ± 11.31 | 5.42 | 123.20 ± 20 | 6.11 | 5.46 |
| BZ | 5.20 ± 1.87 | > 64 | >12.3 | - | - | - | - | - | - |
| MIL | - | - | - | 1.10 ± 0.16 | 58.46 ± 5.37 | 53.14 | - | - | - |
| DOX | - | 0.69± 0.08 | - | - | 0.97± 0.31 | - | 0.24 ± 0.06 | - | - |

SI, selectivity index; BZ, reference drug for Chagas disease; MIL, reference drug for leishmaniasis; DOX, positive control for cytotoxicity evaluation; SI$^1$ CC$_{50}$ HFF-1/IC$_{50}$ *T. cruzi*; SI$^2$ CC$_{50}$ THP-1/IC$_{50}$ *L. infantum;* SI$^3$ CC$_{50}$ HepG2/IC$_{50}$ *T. cruzi*; and SI$^4$ CC$_{50}$ HepG2/IC$_{50}$ *L. infantum*.

for *L. infantum*. The cytotoxicity was observed in host macrophages, fibroblasts and hepatocytes, and the results are shown in Table 1.

## Antiphagocytic effect

The potential for cytochalasins to inhibit *L. infantum* phagocytosis by THP-1 cells was evaluated. The results in Fig 2A show that cytochalasins A-D were able to inhibit phagocytosis on THP-1 cells at all tested concentrations, confirming the preliminary observations [14]. An even greater decrease in infected cells can be observed in Fig 2B, because the cytochalasins were not removed before adding *L. infantum* promastigotes. The results in Fig 2C indicated that cytochalasins A and C, in addition to inhibiting phagocytosis, were toxic to *L. infantum* promastigote parasites.

## Molecular modelling of cytochalasins A-D and actin

Molecular docking studies were performed to underline the potential interactions of the studied compounds with actin within the cytochalasin binding site. The PDB structure of

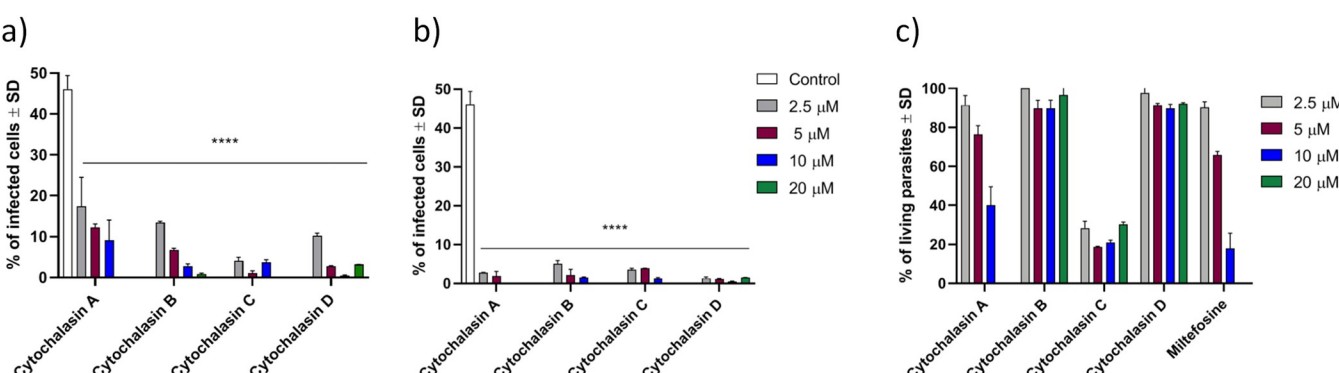

**Fig 2. Ratio of THP-1 cell infection by *L. infantum* promastigotes in the absence (negative control) or presence of cytochalasins A-D (20 to 2.5 μM, 2 h incubation).** A) After 2 hours of incubation, the cytochalasin was removed, and late-stage *L. infantum* promastigotes were added. B) Cytochalasins were not removed after incubation, and late-stage *L. infantum* promastigotes were added. Phagocytosis inhibition is expressed as a percentage of cells infected after counting 200 random cells. C) The toxicity of cytochalasins to promastigotes is expressed as the remaining living parasites after 24 h of incubation. **** p < 0.0001.

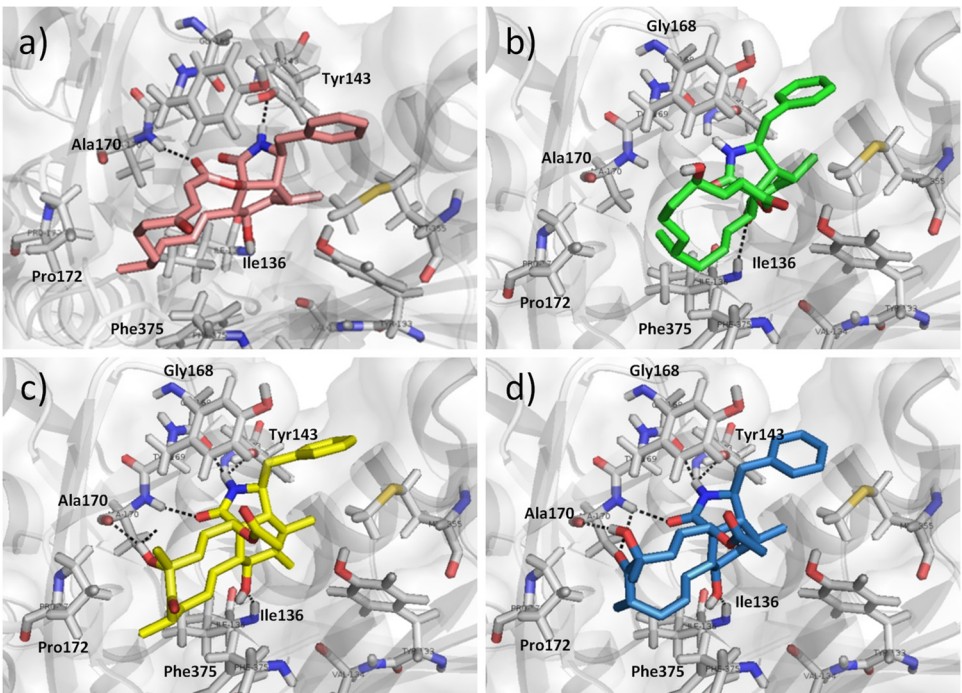

**Fig 3. Intermolecular interactions indicated as black dashed lines for cytochalasins A-D in the binding site for cytochalasin D in actin (cartoon and surface representations as grey, PDB ID: 3EKS).** A) Structure of cytochalasin A (light pink) highlighting intermolecular interactions of 7-hydroxy and backbone nitrogen of Ile136, N-H of the isoindolone ring with hydroxyl of Tyr143, carboxyl of the macrocycle lactone with backbone of Ala170; B) cytochalasin B (green) and the interaction of 7-hydroxy and backbone nitrogen of Ile136, N-H of the isoindolone ring with the backbone carbonyl of Gly168, carboxyl group of isoindolone ring with backbone of Ala170, C) cytochalasin C (yellow) interactions of 7-hydroxy and backbone nitrogen of Ile136, N-H of the isoindolone ring with hydroxyl of Tyr143, carboxyl of the macrocycle lactone with backbone of Ala170 and Gly168, carboxyl group of isoindolone ring and hydroxyl of macrocycle with backbone of Ala170, and D) cytochalasin D (blue) interacts similarly to cytochalasin C and there is an additional interaction of the macrocycle carbonyl with backbone of Ala170.

cytochalasin D crystallized with actin was employed (PDB ID: 3EKS) in these analyses. The predicted binding modes of cytochalasins A-D are shown in Fig 3A–3D. The complexes revealed intermolecular interactions similar to those of the crystallographic complex [18]. The hydrogen bonds of the hydroxyl at position 7 of the ring with the backbone nitrogen of Ile136 were observed for all four cytochalasins. The N-H of the isoindolone ring interacts with either the hydroxyl of Tyr143 or the backbone carbonyl of Gly168. Both the backbone nitrogen and carbonyl of Ala170 interact with the hydroxyl or lactone polar groups of the cytochalasin macrocyclic ring. Considering that the macrocyclic ring of cytochalasins is rather hydrophobic, it interacts with the hydrophobic residues Ile136, Pro172, Phe375, and Ala170.

### Physico-chemical and pharmacokinetic analysis

The experimentally determined Log*D* values for cytochalasins A-D were in the range between 3 and 4 (Table 2), showing moderate to high lipophilicity (moderate Log*D* (0–3); high Log*D* (> 5)). In silico predictions provided lipophilicity values in the range of 2.5 < log*P* < 3.22 (Table 2), in accordance with the in vitro data. During the cell assays, no solubility problems were detected. The kinetic measurements of solubility for cytochalasins A-D in PBS at pH 2.0 and 7.4 were medium to high (Table 2) and were in accordance with the predicted Log*S*. Under physiological conditions (pH 7.4, the same used in the cell assays), the solubility was

**Table 2. ADME properties of cytochalasins A–D determined in vitro and predicted in silico.**

| | Log*D* 7.4 (Oct/buff) | Predicted Log*D* Swiss ADME [20] | Kinetic Solubility pH 2.0 (Log*S*) | Kinetic Solubility pH 7.4 (Log*S*) | Predicted Log*S* (ESOL) [20] | PAMPA permeability Mean Pe ($10^{-6}$ cm/s) | Human liver microsome (HLM) CL$_{int}$(mic) (µL/min/mg) | Mouse liver microsome (MsLM) CL$_{int}$(mic) (µL/min/mg) |
|---|---|---|---|---|---|---|---|---|
| **Cytochalasin A** | 3.907 ± 0.014 | 3.17 | -4.40 | -4.16 | -4.86 | 27.14 ± 2.37 | 7.86 ± 2.05 | 166.93 ± 9.12 |
| **Cytochalasin B** | 3.361 ± 0.008 | 3.22 | -4.11 | -4.11 | -4.93 | 15.78 ± 0.31 | 217.06 ± 6.05 | 62,93 ± 12,58 |
| **Cytochalasin C** | 3.434 ± 0.003 | 2.72 | -4.77 | -4.33 | -4.21 | 14.79 ± 3.28 | 144.40 ± 15.84 | 216,66 ± 18.77 |
| **Cytochalasin D** | 3.043 ± 0.004 | 2.50 | -4.37 | -4.33 | -4.58 | 8.74 ± 0.64 | 188.93 ± 17,53 | 162.53 ± 22.43 |

Classification Criteria: Kinetic Solubility Log*S* < 4.22 → Low solubility; 4.22 < Log*S* < 5 → Moderate solubility; Log*S* > 5 → High solubility [24]. PAMPA permeability Pe > 1.5 * 10–6 cm/s → high permeability; Pe < 1.5 * 10–6 cm/s → low permeability. Clearance potential classification CL$_{int}$ < 25.0 µL/min/mg → Low clearance; 25.0 < CL$_{int}$ < 100.0 µL/min/mg → Moderate clearance; and CL$_{int}$ > 100.0 µL/min/mg → High clearance [25].

similar among the compounds and similarly observed at pH 2.0. The results of the PAMPA assay showed that the permeability of cytochalasins is considerably high (Table 2).

The predicted bioavailability radar in Fig 4 provides a general glance at the compound's drug-likeness, in which the background red-coloured area represents the optimal range for the estimated properties. The predicted properties were in a good range for lipophilicity (LIPO), molecular mass (SIZE), polarity (POLAR), fraction of Csp3 (INSATU), number of rotatable bonds (FLEX), and moderate solubility (INSOLU).

## Discussion

Cytochalasins A-D showed significant potency against the intracellular amastigote forms of *T. cruzi* and *L. infantum*. The potency of cytochalasin A is comparable to that of the standard drug benznidazole (BZ, IC$_{50}$ [T. cruzi] of 5.20 µM) and miltefosine (MIL, IC$_{50}$ [L. infantum] of 1.10 µM). A lower cytotoxicity for host cells when compared to parasite cells was observed, which is also a good feature for this class of natural products. Cytochalasins A-D were more than 10-fold less toxic than the standard control doxorubicin. Structure-activity relationship studies explored the structural and physicochemical requirements related to the antiparasitic effects of these compounds. The presence of the 1,4 diketone (C-20 carbonyl), 14-member macrocyclic ring, and the double bond at C6-C12 was an important feature driving the strong potency exhibited by cytochalasin A. The isoindolone, hydroxyl at position C-7, and phenyl at position C-3 are important for the cytotoxic activity of cytochalasins, and the macrocyclic ring was previously regarded as important for bioactivity [26].

The antiphagocytic properties of cytochalasins were demonstrated in this work and are an interesting strategy for leishmaniasis drug discovery. Further from the phenotypic study, we evaluated the interaction of these compounds with actin. This protein is involved in the internalization of *Leishmania* by host cells, and although the actin of the parasite itself is not inhibited, the inhibition of host cell actin is an interesting strategy to hinder parasite infection.

The ADME (administration, distribution, metabolism and excretion) properties for cytochalasins A-D were assessed using both in silico and in vitro methods. The PAMPA showed that the permeability of cytochalasins is considerably high, which is an important feature for good bioavailability and an important factor for passive diffusion and gastrointestinal absorption. The metabolic stability results assessed by a test in human and mouse liver microsomes indicated that cytochalasin A is more stable than cytochalasins B-D. The predicted

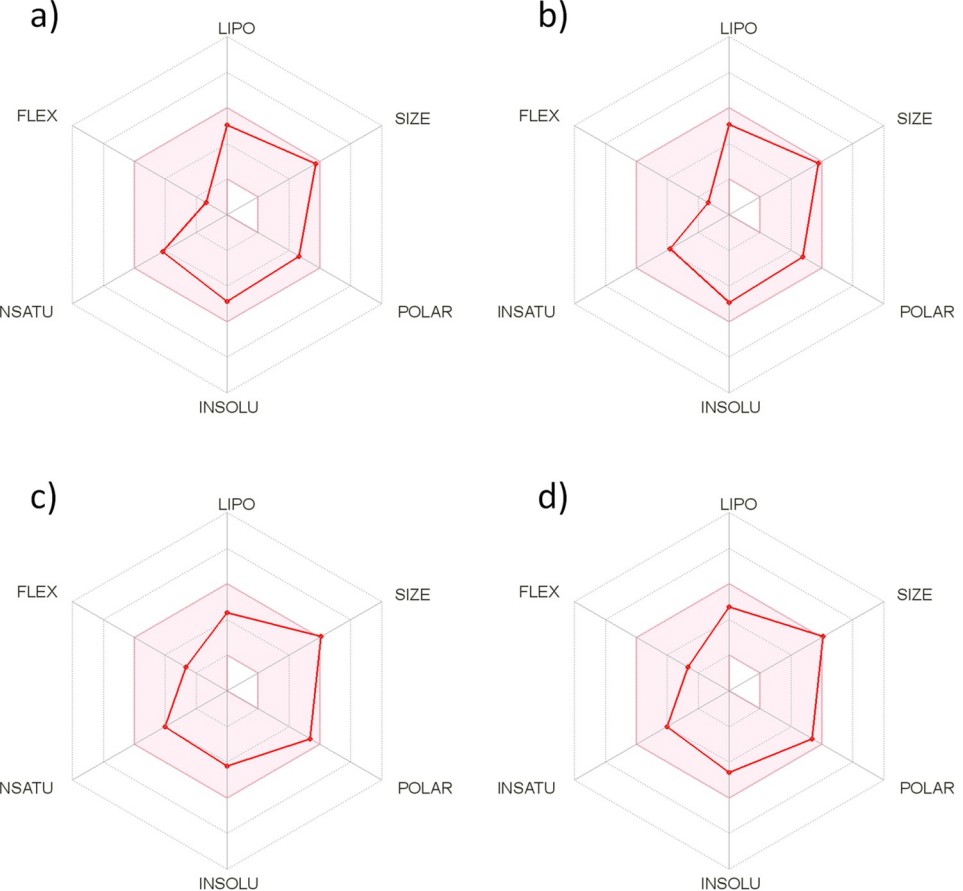

**Fig 4.** Cytochalasins A-D (panels a-d) Bioavailability radar generated using the Swiss ADME web tool. The coloured area represents the optimal range for calculated properties, LIPO (-0.7 < Log$P$ < 5), SIZE (150 g/mol < molecular mass < 500 g/mol), POLAR (20 Å$^2$< TPSA < 130 Å$^2$), INSOLU (0 < Log$S$ < 6), INSATU (0.25 < fraction of Csp3 < 1), and FLEX (0 < nRotb < 9). All the compounds show good predicted oral bioavailability.

bioavailability radar is useful for a general glance at the compound's drug-likeness, and the results indicate good oral bioavailability for cytochalasins A-D.

Our strategy was to screen a natural product library from Brazilian organisms that is an important source of chemical diversity for developing drugs to treat neglected diseases. This work led to the identification of cytochalasins A-D, natural products presenting antileishmanial and antiphagocytic activity against *L. infantum* with good ADME parameters, and these products consisted of promising lead compounds for antiparasitic drug discovery. Similarly, other natural products could be identified in a variety of species to treat a diversity of diseases. Lastly, these results provide further advancements for the understanding of the cytochalasins' anti-parasitic activity, adding useful information on the drug discovery of new antiparasitic agents.

## Acknowledgments

The authors acknowledge the Nuclei of Bioassays, Biosynthesis and Ecophysiology of Natural Products (NuBBE), Department of Organic Chemistry, Institute of Chemistry, São Paulo State University (UNESP) for providing samples for the screening assays.

## Author Contributions

**Conceptualization:** Marilia Valli, Adriano Defini Andricopulo.

**Funding acquisition:** Vanderlan da Silva Bolzani.

**Investigation:** Marilia Valli, Julia Medeiros Souza, Rafael Consolin Chelucci.

**Methodology:** Marilia Valli, Julia Medeiros Souza, Rafael Consolin Chelucci, Carolina Rabal Biasetto.

**Project administration:** Vanderlan da Silva Bolzani, Adriano Defini Andricopulo.

**Resources:** Carolina Rabal Biasetto, Angela Regina Araujo.

**Supervision:** Angela Regina Araujo, Adriano Defini Andricopulo.

**Writing – original draft:** Marilia Valli, Julia Medeiros Souza, Rafael Consolin Chelucci, Carolina Rabal Biasetto.

**Writing – review & editing:** Marilia Valli, Angela Regina Araujo, Vanderlan da Silva Bolzani, Adriano Defini Andricopulo.

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
