## [Decision Letter · Decision Letter 0]

9 Sep 2022

Identification of natural cytochalasins as leads for neglected tropical diseases drug discovery

PONE-D-22-16784

Dear Dr. Valli,

We’re pleased to inform you that your manuscript has been judged scientifically suitable for publication and will be formally accepted for publication once it meets all outstanding technical requirements.

Kind regards,

Bhaskar Saha

Academic Editor

PLOS ONE

   "This work was supported by the Fundação de Amparo à Pesquisa do Estado de São Paulo (FAPESP, https://fapesp.br/) grants #2013/07600-3 (CIBFar-CEPID), #2014/50926-0 (INCT BioNatCNPq/FAPESP), Conselho Nacional de Desenvolvimento Científico e Tecnológico (CNPq, https://www.gov.br/cnpq/pt-br) and Coordenação de Aperfeiçoamento de Pessoal de Nível Superior (CAPES, https://www.gov.br/capes/pt-br) for grant support and research fellowships. The authors acknowledge the scholarships conferred to MV (Fapesp #2019/05967-3) and JM (Fapesp #2019/06034-0)."

Please respond by return e-mail so that we can amend your financial disclosure and competing interests on your behalf.

Additional Editor Comments (optional):

Reviewers' comments:

Reviewer's Responses to Questions

**Comments to the Author**

1. Is the manuscript technically sound, and do the data support the conclusions?

Reviewer #1: Yes

Reviewer #2: Yes

2. Has the statistical analysis been performed appropriately and rigorously? 

Reviewer #1: Yes

Reviewer #2: Yes

3. Have the authors made all data underlying the findings in their manuscript fully available?

Reviewer #1: Yes

Reviewer #2: Yes

4. Is the manuscript presented in an intelligible fashion and written in standard English?

Reviewer #1: Yes

Reviewer #2: Yes

5. Review Comments to the Author

Reviewer #1: Authors have studied anti-protozoan drug activity and bio-availability of cytochalasins suggesting that cytochalasins can be potentially used as drugs to treat infections caused by Trypanosoma and Leishmania.

Reviewer #2: The manuscript entitled “Identification of natural cytochalasins as leads for neglected tropical diseases drug discovery” is an interesting study. Authors in this manuscript using combination of computational and biological approaches identified and characterized four cytochalasins as drug candidate for neglected tropical diseases. Data shown in this manuscript are convincing.

---

## [Editor Report · Acceptance letter]

16 Sep 2022

PONE-D-22-16784 

Identification of natural cytochalasins as leads for neglected tropical diseases drug discovery 

Dear Dr. Valli:

I'm pleased to inform you that your manuscript has been deemed suitable for publication in PLOS ONE. Congratulations! Your manuscript is now with our production department. 

Kind regards, 

on behalf of

Dr. Bhaskar Saha 

Academic Editor

PLOS ONE